# Information Available to Parents Seeking Education about Infant Play, Milestones, and Development from Popular Sources

**DOI:** 10.3390/bs13050429

**Published:** 2023-05-19

**Authors:** Julie M. Orlando, Andrea B. Cunha, Zainab Alghamdi, Michele A. Lobo

**Affiliations:** 1Biomechanics & Movement Science Program, University of Delaware, Newark, DE 19713, USA; jorlando@udel.edu (J.M.O.); zainabgh@udel.edu (Z.A.); 2Physical Therapy Department, Munroe Meyer Institute, University of Nebraska Medical Center, Omaha, NE 68106, USA; abaraldicunha@unmc.edu; 3Physical Therapy Department, University of Delaware, Newark, DE 19713, USA

**Keywords:** child development, internet, infancy, parenting practices, play and playthings, health education, information seeking, content analysis, milestones

## Abstract

Parents commonly seek information about infant development and play, yet it is unclear what information parents find when looking in popular sources. Play, Milestone, and Development Searches in Google identified 313 sources for content analysis by trained researchers using a standardized coding scheme. Sources included websites, books, and apps created by professional organizations, commercial entities, individuals, the popular press, and government organizations/agencies. The results showed that for popular sources: (1) author information (i.e., qualifications, credentials, education/experience) is not consistently provided, nor is information about the developmental process, parents’ role in development, or determining an infant’s readiness to play; (2) milestones comprise a majority of the content overall; (3) search terminology impacts the information parents receive; (4) sources from the Milestone and Development Searches emphasized a passive approach of observing developmental milestones rather than suggesting activities to actively facilitate learning and milestone development. These findings highlight the need to discuss parents’ online information-gathering process and findings. They also highlight the need for innovative universal parent-education programs that focus on activities to facilitate early development. This type of education has potential to benefit all families, with particular benefits for families with children who have unidentified or untreated developmental delays.

## 1. Introduction

A key problem in pediatric rehabilitation is the lack of provision of early intervention (EI) services to young children requiring those services to mitigate developmental delays. Less than 10% of eligible children receive EI services in the United States [1,2]. Traditional approaches to address this challenge include working within the medical and EI systems to improve surveillance, screening, assessment tools, and procedures, as well as addressing barriers to service provision [3,4,5,6,7]. In parallel, we propose that an innovative and effective way to address this challenge would be through the development of high-quality universal education programs that teach parents how to engage and support infants in ways shown to promote learning and development. Universal parent education programs that provide information via the sources parents prefer to access have the potential to positively impact parental knowledge [8], parent–child interaction [9,10], child development [11], and discussions between parents and care providers [12]. These outcomes would be beneficial for all children, while being especially beneficial for children with unidentified and/or untreated developmental delays. This study analyzes the existing content universally available to parents in popular sources to identify what information is already being shared with parents and to determine whether this information aligns with current developmental science.

Parental knowledge about infant development was originally defined as a “parent’s understanding of developmental norms and milestones, processes of child development, and familiarity with caregiving skills” (p. 1187, [13]). This definition continues to be used today [14,15]. Parental knowledge is positively related to parent–child interaction [16,17,18,19] and infant development [13,14,20]. These relations have been demonstrated among parents of differing ages [14], varying socioeconomic status [16], across cultures [15,21,22], and with infants with typical development as well as those born preterm and at increased risk for developmental delay [20]. Importantly, there is the potential to improve parental knowledge [8], parent–child interaction [10,23,24], and infant development [11,25,26] through parent education interventions [27,28].

In contrast to developmental knowledge, parental knowledge about infant play has not been as thoroughly studied. Damast, Tamis-LeMonda, and Bornstein determined that mothers who had greater knowledge about play were more likely to offer higher-level play activities for their young children, potentially leading to advanced development [29]. Parents who participated in an early positioning and handling education program with their infants had infants who demonstrated short-term advances in prone skills and longer term advancements in crawling, standing, and walking compared to a control group [11]. Similarly, educating parents to encourage infants’ general arm movements (i.e., using wrist tethers to control toys) advanced the ability to reach for objects in two-month-olds with typical development [30]. Thus, parental knowledge about play and the information that parents receive about how to interact with their infants can impact parent–child interaction and infant development.

Parents and pediatric clinicians actively seek information about infant development and play [31]. In a national survey of 2200 parents conducted by Zero to Three^®^, most parents agreed that new research about child development could improve their parenting; about half of the parents wanted more information about how to be a better parent and/or wished they had known more about brain development when their child was younger [32]. Pediatricians and other pediatric clinicians often recommend resources about infant development and play to parents [12,31,33,34]. In a survey of 112 parents of children under two years of age, a majority of parents reported searching for information about development (88.4%) or play (68.8%). Similarly, a majority (92.8%) of the 138 EI clinicians surveyed reported recommending resources about infant development and play to parents [31]. Importantly, parents exposed to the Learn the Signs Act Early Campaign by the Centers for Disease Control (CDC), a program that aims to help families learn about and participate in developmental surveillance and improve early identification of delays [35], had a greater understanding of milestones and engaged in more discussion about development during pediatrician visits [12,36]. These findings suggest that the education that parents receive about development and play may also impact the way that parents engage with healthcare providers, which may impact the early identification of delays.

Parents around the world have reported using the internet and other mass media to find information about infant development [31,37,38,39]. In a systematic review, Kubb et al. identified that there is a high prevalence of parents using the internet to find health information, including information about specific health conditions, treatment options, and general health information about their children [40]. Notably, the authors describe that “Google was the most common starting point for general health information” (p. 177, [40]). In addition to internet searches, parents also turn to books and mobile applications (apps) for parenting advice [32] and information about infant development and play [31]. It is important for pediatric clinicians to understand the content that parents encounter when searching for information about infant development and play.

Content analyses provide insight into the resources that are available for a target population and can provide recommendations for improved future content creation. For instance, to determine if the needs of parents of children with cerebral palsy were being appropriately addressed, Lau S.K. examined the content on cerebral palsy agency websites and concluded that the agency websites were meeting the needs of parents [41]. Only two content analyses have evaluated the information available to parents regarding infant development, and none have evaluated content regarding infant play. Williams et al. analyzed sources discovered after searching Google and Yahoo search engines for information about child development, parenting, and developmental milestones [42]. The authors reviewed 44 websites and described the accuracy of their content in comparison to the American Academy of Pediatrics’ book, *From Birth to Five Years*. They concluded that many of the resources reviewed were accurate, yet they often lacked clarity, were incomplete, or were difficult to navigate. Dewitt et al. evaluated the functionality and content of apps about development from birth through 5 years [43]. They found that few app development teams described the inclusion of a subject matter expert, and only 15% of the apps included content about developmental milestones. Overall, little is known about the information parents may encounter when they look to popular sources for information about infant development and play. This study aimed to fill that knowledge gap by systematically evaluating the content of popular sources about infant play, milestones, and development available to parents searching in the United States (US). Specifically, we aimed to describe: (1) the types of sources available and the authors of these materials; (2) the source content (i.e., number of play activities, milestones, and toys suggested); (3) the information shared with parents regarding developmental processes, the role that parents play in development, and how to determine when an infant is ready to play. These data were collected to describe the information available to parents as well as to identify whether the information presented emphasizes infants’ daily experiences and environment in a manner that aligns with current developmental theories. An understanding of the educational materials currently available is critical to evaluate the need for and to inform the development of early parent education programs that can serve as innovative, universal rehabilitation tools to benefit all children with specific benefits for children with or at risk for developmental delays.

## 2. Materials and Methods

### 2.1. Source Selection

Searches were conducted using the Google search engine between October 2019 and March 2021 with the following search phrases: (1) “How to play with baby” (i.e., referred to in this paper as the Play Search); (2) “Baby milestones first year” (i.e., the Milestone Search); or (3) “Your baby’s development first year” (i.e., the Development Search). Each search was conducted one time during this period. The search terms were selected to replicate searches that a parent may perform. The search results included videos, websites, books, and apps and were saved and exported as a CSV file using the SEOquake plugin (Boston, MA, USA) for Google Chrome. The top 150 results from each search were screened based on the inclusion and exclusion criteria described below. We selected 150 as our cut-off for screening based on the decreased relevance of the search results between 100 and 150. Sources were included in the content analysis if they: (1) involved infants within the 0–12-month age range; (2) were written in English; (3) included content related to infant development and/or play (examples of sources that did not meet this inclusion criterion were keepsake items, growth charts, and sources with only feeding or nutrition content). Sources were excluded if they: (1) were not accessible; or (2) only served a commercial purpose (i.e., promoting a product). Content for each source that met the screening criteria was archived at the time of screening for future analysis. In addition, sources that parents reported referencing for information about infant development and play from an online survey *(n* = 112 participants) conducted by the authors were also screened [31]. Parents were specifically asked to list examples of sources they had accessed for information about infant development or play. We included these sources in our screening process to ensure that our content analysis broadly represented sources accessed by parents.

Our initial search (between October 2019 and March 2021) resulted in 268 sources. To ensure the sample reflected current information, the content for these sources was reviewed in November 2022, and recoding was performed in cases where a new edition was published for a book or where content had been altered for a website. We found that only 19 sources (7.06% of the total number of sources) had modified content; 25 sources (9.33% of the total) had updated the date the source was last reviewed without changing any of the relevant content. The updated coding for the 19 sources with modified content resulted in changes to only 3.6% of the play activities recommended, 2.47% of the milestones listed, and 1.61% of the toys recommended, suggesting that this content remains relatively stable across time. To further update the dataset, we also repeated the Play, Milestone, and Development Searches in September of 2022. We screened the top 50 results from each search for inclusion in order to identify examples of the most relevant additional sources. We added an additional 22 Play, 8 Milestone, and 14 Development sources (44 total) to the content analysis from this more recent search.

### 2.2. Content Coding Procedures

The codebook was developed by a team of experts in child development and early intervention (Appendix A). It aimed to gather information about each source, including information about the authors and about the depth and type of information within each source. The *source type* was classified as book, website, or app. The *author type* was classified as commercial entity, government organization/agency, individual, popular press, or professional organization. *Author credentials* were coded as being present if there was a description of the degree(s) (e.g., PhD, MD, RN, LSW), certifications, or licenses earned by the author. *Author qualifications* were coded as being present if there was a description about the author’s experience or expertise, including being a parent. Although sources from professional organizations did not often identify the authors of their content, we credited them as having provided credentials and qualifications based on the noted affiliation with the professional organization. *Author education/experience* was coded based on descriptions provided within the sources as falling into the categories of early childhood education, healthcare, human services (e.g., social worker, therapist, psychologist), parental experience, other (e.g., media relations, editor), or unspecified (e.g., listing “Dr.” without further description); multiple selections were possible.

The depth (i.e., amount of content) and type of information within each source was comprised of the inclusion (1 = yes, 0 = no) of play activities, milestones, and toy recommendations as well as the quantity (i.e., the count of all of the play activities, milestones, and toy recommendations described within the source). A play activity was defined as something that a person could do with an infant or something that the infant could do on their own (e.g., hold your baby and dance to their favorite song; give your baby a container filled with scarves and toys and let them explore; talk with your baby). If multiple activities were described within one sentence, each activity was coded separately. A milestone was defined as a behavior that an infant would be expected to perform by a specified timepoint (e.g., your baby will sit without support by nine months). A toy recommendation was coded when an object was recommended to parents for use by the infant or parent (e.g., rattles, spoons, plastic water bottles, scarves, books).

We also coded whether the source shared information about: (1) developmental processes (i.e., information describing how development happens, such as developmental theories or factors influencing development); (2) a parent’s role in development (i.e., information describing the role that parents can play in impacting developmental outcomes for their infants); (3) how to determine if an infant is ready to play (i.e., directing parents to infants’ signs of readiness to play signs or signs of overstimulation). Any text related to these topics was copied into the coding files. Text content related to the developmental process, parents’ role in development, and infants’ readiness to play was independently evaluated for themes by two researchers. The researchers then converged to review, discuss, and come to agreement about the emergent themes from this coding.

The content for each source was coded in Google Sheets by research assistants trained to reach greater than 90% inter-rater agreement. The content within each source was coded by two independent coders. The primary coder first extracted and coded all relevant information from the source. The secondary coder then coded the relevant material from the same source and added any material that may have been missed. The primary and secondary coding sheets were compared using a custom MATLAB program (Natick, MA, USA). Only 8.28% of the data were found to disagree. All disagreements were reviewed by the first, second, or third author, all of whom were involved in developing the coding scheme or a senior research assistant with additional training. The review involved returning to the source content to ensure the accuracy and completeness of the dataset. The final data (Appendix A) were then compiled and stored in a custom Claris FileMaker Pro (Cupertino, CA, USA) relational database.

### 2.3. Statistical Analysis

Data were analyzed using descriptive statistics, including frequencies and percentages. The source medium, author type, author credentials, and author qualifications were described as a percentage of the total number of unique sources (i.e., eliminating redundancies in cases where a source was encountered through more than one search) in order to describe the materials available to parents as a whole. In contrast, since one purpose of this study was to characterize the information parents would encounter from each type of search (i.e., Play, Milestone, or Development), when analyzing author education/experience and outcomes related to depth and type of information, data from individual sources that were found through more than one search type (*n* = 47, 15.01% of all sources) were included in the analyses for each of their associated searches.

Statistical analyses were conducted using SPSS version 29 (Armonk, NY, USA). To determine whether author characteristics and information shared varied based on the search type (i.e., Play, Milestone, or Development), Pearson’s Chi-Square Test of Independence was used to evaluate whether there were relations between search type and: (1) author type; (2) author credentials; (3) author qualifications; (4) the inclusion of source content (i.e., play activities, milestones, and toy recommendations); (5) the inclusion of information related to developmental process, parent’s role in development, and readiness to play; (6) themes identified related to developmental process, parent’s role in development, and infant readiness to play. To determine whether the authors of the content from the Play, Milestone, or Development Searches differed, relations among author education/experience and the type of search conducted were evaluated with Fisher’s Exact Test. Author education/experience was a multiple selection variable (e.g., parental experience and healthcare). Only author education/experience combinations with greater than five responses in total among the Play, Milestone, and Development Searches were included in analyses. Fisher’s Exact Test accounts for instances when the observed frequency of an author education/experience variable was greater than five overall but less than five within a specific search (i.e., Play, Milestone, or Development). The significance level was set to alpha = 0.05. Post-hoc analyses were conducted using standardized adjusted residuals and Bonferroni adjustments with the null hypothesis that all observed values were equally distributed [44,45,46].

## 3. Results

The Play Search resulted in 122 sources published between 1995 and 2022 (median: 2019). The Milestone Search resulted in 126 sources published between 2005 and 2022 (median: 2019). The Development Search resulted in 112 sources published between 2000 and 2022 (median: 2020). In total, 313 unique sources were analyzed; 47 sources were found through more than one search. Examples of the sources reviewed included books, such as *Your Baby’s First Year* by the American Academy of Pediatrics [47] and *What to Expect the First Year* by Heidi Murkoff [48]; websites, such as Pathways.org^®^ [49], ZerotoThree.org^®^ [50], BabyCenter.com^®^ [51] (accessed on 5 April 2023), and Learn The Signs Act Early by the Centers for Disease Control and Prevention [52]; and apps, such as BabySparks^®^ [53] and Kinedu^®^ [54]. The complete list of sources reviewed is available within the final data (Appendix A).

### 3.1. Description of the Sources

#### 3.1.1. Source Medium and Author Type

The majority of the sources (*n* = 289, 92.33%) were websites which may be a reflection of the search process being conducted online. However, books (*n* = 14, 4.47%) and apps (*n* = 10, 3.19%) discovered through online searches and/or through the parent survey were also included. Of the 313 unique sources, most were created by professional organizations (*n* = 176, 56.23%), followed by commercial entities (*n* = 50, 15.97%), individuals (*n* = 42, 13.42%), the popular press (*n* = 33, 10.54%), and government organizations/agencies (*n* = 12, 3.83%). Within each search type, most of the sources were from professional organizations (Play: *n* = 51, 41.80%; Milestone: *n* = 85, 67.46%, Development: *n* = 76, 67.86%). There was a significant relation between the type of search conducted and the author type (X^2^(8) = 38.77, *p* < 0.001). Individual authors were significantly more likely to create content discovered from the Play Search (Z = 4.97, *p* < 0.001) while professional organizations were significantly less likely to create content discovered from the Play Search (Z = −4.72, *p* < 0.001).

#### 3.1.2. Author Credentials, Qualifications, and Education

Of the 313 unique sources, author credentials (*n* = 213, 67.73%) and author qualifications (*n* = 225, 71.88%) were often included. There was a significant relation between the search type and the presence of author credentials (X^2^(2) = 11.75, *p* = 0.003) and qualifications (X^2^(2) = 47.30, *p* < 0.001). Author credentials were significantly less likely to be present in the sources discovered in the Play Search (Z = −3.41, *p* < 0.001). Author qualifications were significantly more likely to be present in the sources from the Milestone (Z = 3.33, *p* < 0.001) and Development Searches (Z = 3.6, *p* < 0.001) and less likely to be present in the sources from the Play Search (Z = −6.87, *p* < 0.001).

Author education/experience was reported by 137 sources (38.02% of the unique sources), and there were 17 combinations of author education/experience selections (Appendix A). *Healthcare* education/experience was most frequently reported (*n* = 54, 39.42%), followed by *healthcare and parental education/experience* (*n* = 15, 10.95%) and *parental experience and other* (*n* = 11, 8.03%). Sources authored by individuals from healthcare were often discovered in the Milestone (*n* = 22, 52.38%) or Development (*n* = 19, 48.72%) Searches. Authors who were described solely as parents were only 5.11% of the sample (*n* = 7) and were most often discovered from the Play Search (*n* = 6, 85.71%). Interestingly, authors who identified as parents with healthcare education/experience were discovered most often from the Play Search (*n* = 10, 66.67% of healthcare and parents). Only healthcare, healthcare/parent, parent/other, early childhood education, human services, parent, parent/human services and “other” received greater than five total responses and were therefore included in the analyses. There was a significant relation between search type and author education; Fishers Exact test (X^2^(14) = 30.09, *p* = 0.002) and post-hoc analyses identified that individuals with healthcare education/experience were less likely to have authored sources from the Play Search (Z = −3.22, *p* = 0.001).

### 3.2. Description of the Source Content

#### 3.2.1. Inclusion of Play Activities, Milestones, and Toy Recommendations

The presence of play activity, milestone, and toy recommendation content was evaluated within each search type. Play activity content was found within 95.90% (*n* = 117) of the sources from the Play Search, 41.27% (*n* = 52) of the sources from the Milestone Search, and 59.82% (*n* = 67) of the sources from the Development Search. There was a significant relation between the presence of play activity content and the search type (X^2^(2) = 85.30, *p* ≤ 0.001). Post-hoc analyses identified that sources discovered from the Play Search were significantly more likely to include play activity content (Z = 8.68, *p* < 0.001), while sources from the Milestone Search were significantly less likely to include play activity content (Z = −7.12, *p* < 0.001).

Milestone content was found within 65.57% (*n* = 80) of the sources from the Play Search, 98.41% (*n* = 124) of the sources from the Milestone Search, and 94.64% (*n* = 106) of the sources from the Development Search. There was a significant relation between the presence of milestone content and the search type (X^2^(2) = 65.78, *p* ≤ 0.001). Post-hoc analyses identified that sources discovered from the Development and Milestone Searches were significantly more likely to include milestone content (Development: Z = 3.15, *p* = 0.002; Milestone: Z = 4.95, *p* < 0.001), while sources from the Play Search were significantly less likely to include milestone content (Z = −8.07, *p* < 0.001).

Toy recommendations were found within 90.16% (*n* = 110) of the sources from the Play Search, 42.86% (*n* = 54) of the sources from the Milestone Search, and 58.93% (*n* = 66) of the sources from the Development Search. There was also a significant relation between the presence of toy recommendations and the search type (X^2^(2) = 61.86, *p* ≤ 0.001). Post-hoc analysis identified that sources discovered from the Play Search were significantly more likely to include toy recommendations (Z = 7.43, *p* < 0.001), while sources from the Milestone Search were significantly less likely to include them (Z = −6.1, *p* < 0.001).

#### 3.2.2. Quantity of Play Activities, Milestones, and Toy Recommendations

The Play Search had a total of 5064 combined items (2254 play activities, 1551 milestones, 1259 toy recommendations), the Milestone Search had 10,085 combined items (1377 play activities, 7898 milestones, 810 toy recommendations), and the Development Search had 8854 combined items (1602 play activities, 5650 milestones, 1602 toy recommendations; Figure 1).

### 3.3. Description of the Content Related to Developmental Process, Parents’ Roles in Development, and Infants’ Readiness to Play

Information about the developmental process was present in 37.70% (*n* = 46) of the sources from the Play Search, 49.21% (*n* = 62) of the sources from the Milestone Search, and 38.39% (*n* = 43) of the sources from the Development Search. There was no significant relation between the presence of information about developmental process and the search type (*p* = 0.122).

Information about parents’ role in development was present in 46.72% (*n* = 57) of the sources from the Play Search, 41.27% (*n* = 52) of the sources from the Milestone Search, and 35.71% (*n* = 40) of the sources from the Development Search. There was no significant relation between the presence of information about parents’ role in the development and the search type (*p* = 0.233).

Information about infant readiness for play was present in 22.13% (*n* = 27) of the sources from the Play Search, 7.94% (*n* = 10) of the sources from the Milestone Search, and 8.93% (*n* = 10) of the sources from the Development Search. There was a significant relation between the presence of information about infant readiness to play and the search type (X^2^(2) = 13.44, *p* = 0.001). Post-hoc analysis identified that sources discovered from the Play Search were significantly more likely to include information about infant readiness for play (Z = 3.66, *p* < 0.001).

The themes identified in the content regarding developmental process, parents’ roles, and infants’ readiness to play can be seen in Figure 2. Among the developmental process themes, three varied in relation to the search type. There was a significant relation between the type of search and the presence of information stating that *milestones occur at their own pace* (X^2^(2) = 35.20, *p* < 0.001). Post-hoc analyses identified that sources from the Milestone Search were more likely to include this theme (Z = 5.61, *p* < 0.001), while sources from the Play Search were less likely to include it (Z = −4.6, *p* < 0.001). There was also a significant relation between the search type and the theme that *infants learn through play* (X^2^(2) = 28.78, *p* < 0.001), which was more likely to be present in sources from the Play Search (Z = 5.31, *p* < 0.001) and less likely to be present in sources from the Milestone Search (Z = −3.43, *p* < 0.001). Further, there was a significant relation between the search type and content about *development occurring in a specific order* (X^2^(2) = 9.75, *p* = 0.008), with post-hoc analyses identifying that sources from the Milestone Search were more likely (Z = 2.63, *p* = 0.009) and sources from the Play Search were less likely (Z = −2.83, *p* = 0.005) to discuss this.

Within the themes related to the parents’ role in development, one varied in relation to search type. There was a significant relation between the search type and the theme that *parents should identify or discuss concerns about development with their child’s primary care provider*; X^2^(2) = 27.26, *p* < 0.001). Sources from the Milestone Search were more likely (Z = 4.77, *p* < 0.001) to include information about the importance of identifying and bringing up concerns with the primary care provider, while sources from the Play Search were less likely (Z = −4.32, *p* < 0.001) to include this information.

## 4. Discussion

There were several interesting findings from this content analysis of the popular sources of information available to parents searching for education about infant play, milestones, and development. Information was shared with parents via a variety of media, including websites, books, and apps. While author credentials or qualifications were provided in most cases, for about a third of the content, it remained unclear who authored the materials and/or what qualified them to do so. Moreover, author’s education or experience was described for just over a third of the sources. Information about the developmental process and about parents’ roles in development was shared in less than half of the sources, and information about determining whether an infant was ready to play or overstimulated was provided in less than a quarter of the sources. Below, we discuss the key findings of the content analysis along with their implications.

An important novel finding from this content analysis is that, overall, parents who perform searches such as the ones conducted here are most likely to learn about milestones when looking for information about infant play, milestones, and development. Milestones outnumbered play activities by about three to one and toy recommendations by about four to one across all of the content analyzed. Developmental science is a relatively new scientific field, and its early decades focused primarily on documenting developmental products—noting which behaviors emerged and the timeline in which they were observed [55,56,57]. More recent decades have been marked by efforts to better understand the processes through which developmental milestones emerge [57,58,59]. Parent education programs have existed in the US for almost as long as the field of developmental science itself. For example, the Parent Teacher Association (PTA) was founded in 1897 and has published a variety of materials to educate parents [60]. The preponderance of milestones in the educational materials available to parents may reflect that the content is delayed in keeping pace with the science of development.

Another novel finding from this study is that parents will likely encounter distinct types of information created by different groups of authors when they perform searches for information specifically about infant play, milestones, or development. When seeking information about infant play using search terms such as those employed in this study, parents are likely to find information about play activities they can engage in with infants and recommendations for toys and common objects to use in play. On the contrary, when seeking information about milestones using search terms such as those employed in this study, parents are likely to encounter lists of developmental milestones infants are expected to achieve. Interestingly, if parents search for information about infant development using search terms such as those employed in this study, the content closely mirrors that found through the Milestone Search. This perpetuates the concept that development is characterized by the milestones that infants achieve rather than the processes involved in supporting the development of those milestones [56]. Historically, infant development has been widely defined as a series of changes in developmental milestones [55,61], but it is now recognized that infant development is a cascading process that is shaped by a variety of factors, including environmental factors and experiences [62]. Current developmental theory and interventions aimed at advancing development acknowledge that factors such as early experiences [63], postural stability [64], and the environment [65] (e.g., the physical environment [66] and parent–child interaction [67]) can serve as drivers for developmental change. With a strong focus on milestones and less focus on activities and objects that can be used to drive the advancement of those milestones, popular sources share information that reflects an outdated view of development and misses the opportunity to educate parents about how to promote development for young children.

The results of this study are the first to our knowledge to highlight that, although some healthcare professionals have developed educational content for parents of young children that is accessible via online searches (i.e., they authored about a quarter of the sources reviewed), the content created by healthcare providers and other authors should be updated to reflect more current, empirically supported developmental theories [62,68,69]. Those with healthcare education/experience were more likely to author content that was found through the Milestone Search. Importantly, this content primarily listed developmental milestones and advised parents to observe for these milestones as they occur at their own pace and in a specific order. This approach reflects more outdated developmental theories, such as neural-maturation theory, that focused on the typical progression of milestone emergence with a primary driver, such as maturation of the nervous system, causing those changes [56]. Further, content from the Milestone Search was less likely to include play activity suggestions, to recommend toys, or to highlight that infants learn through play. Therefore, like more outdated developmental theories, this content placed less emphasis on the infant’s environment and daily experiences. One potential benefit of this content was that it was more likely to advise parents to communicate with their primary care providers should they note delays in the emergence of developmental milestones. This may support the early identification of delays and utilization of EI services, as children whose parents have developmental concerns may be more likely to receive an EI evaluation and be identified as eligible for services compared to children whose parents who did not have developmental concerns [70].

An innovative approach for healthcare providers and child development experts to optimize development for young children may be through the creation of universal parent education materials that reflect current developmental science. Interestingly, in the current study, most authors who identified as healthcare providers alone published content found through the Milestone and Development Searches, while most authors who identified as healthcare providers and parents published content found through the Play Search. Furthermore, parents and other individuals, rather than professional organizations, were more likely to author content found through the Play Search. This suggests that parents as a whole, including those with healthcare education/experience, recognize the value of and the need for educational content related to infant play. It also suggests that the importance of play and daily activity may not be understood by the professional organizations currently engaged in educating parents. Most parents reportedly seek information about how to play with their infants [31]. Current empirically supported developmental theories, including ecological systems theory and dynamic systems theory, emphasize the critical role of experiences in constructing and shaping children’s developmental trajectories. They claim that milestones do not emerge due to time or maturation, rather they are learned through children’s ongoing, daily experiences [71,72,73]. With parents eager to learn about how to play with their infants and with daily parent–child activity serving as the foundation for milestone development, there is a prime opportunity for healthcare providers and child development experts to generate high-quality educational content for parents with the aim of advancing children’s developmental trajectories [74]. This content should teach parents ways to shape the environment and to enhance learning opportunities for infants [11,75,76].

The results of this study can help pediatric healthcare providers and educators understand what information parents likely encounter through popular sources. The results also highlight the need for updated educational materials for parents. Professionals should engage in discussions with parents about their online information search processes and results. Professionals may direct parents to utilize different search terms to better direct them to the type of information they desire. For example, if parents are interested in learning about play activities or toys, replicating the Play Search would be most effective. If parents are interested in learning about milestones, this content was readily available within all of the searches, however, replicating the Milestone Search would produce the highest volume of milestone content. Future research should further explore the content parents receive when they use a greater variety of search terminology.

The results should be Interpreted”Iit’ a consideration of the study’s limitations. One limitation may be that the majority of sources analyzed were found online. While this likely reflects the search practices of most parents in the US [40], it may not reflect the search practices of all. In addition, the searches were conducted in one location in the US and were limited to English. Geographic location can impact search engine optimization and therefore, future research may benefit from a broader search region. Future research should expand the search locations and compare the results of searches conducted in different countries and languages. Another limitation is that the study reports only on the quantity of play activities, milestones, and toy recommendations shared with parents. It remains unclear whether directing parents to this information would benefit an infant’s development because the quality of the information was not evaluated. Future research should evaluate the quality of the information shared with parents to identify high-quality sources.

The clinical implications of this study are that current, accessible, popular sources available to parents emphasize developmental products (i.e., the milestones that an infant should achieve) and a more passive approach to development. When seeking information about milestones or development, parents are more likely to find a large amount of information listing expected milestones along with instructions to wait for the milestones to emerge and to alert healthcare providers if a developmental concern is noted. Less than a quarter of the content suggested interactions with infants and objects to facilitate infants’ development. In contrast, when seeking information about how to play with infants, parents are likely to find a greater number of activity and toy suggestions along with an emphasis on the importance of parents’ role in development to support early learning. It is important for clinicians who interact with parents of infants to critically review and understand the available resources in order to counsel parents about information seeking practices. Additionally, professionals should engage in discussions with parents about how to critically review the content they access. Professionals should also familiarize themselves with the popular content in order to direct parents to sources that will effectively meet their educational needs. Furthermore, rehabilitation and child development experts should be developing innovative approaches to meet parents’ educational needs that better align with current developmental science. This could be achieved through consultation to improve existing resources along with the creation of novel websites, books, and/or apps geared towards parents. Specifically, universal parent education programs that highlight not only which milestones are expected in development, but also the activities that parents can implement and objects they can use to advance the development of those milestones might serve as effective early intervention tools.

## Figures and Tables

**Figure 1 behavsci-13-00429-f001:**
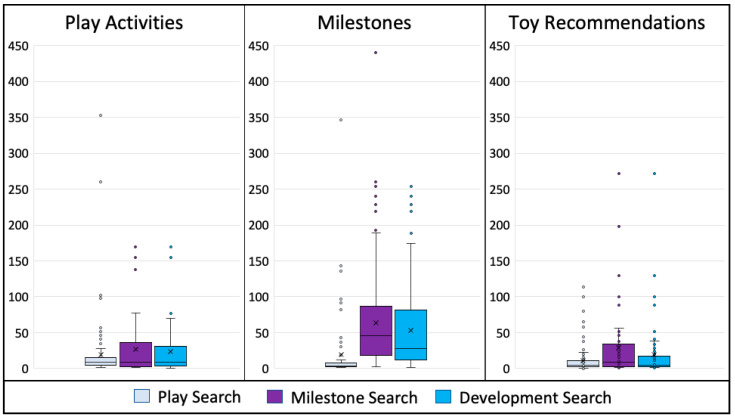
Number of play activities, milestones, and toy recommendations identified in the content from the Play, Milestone, and Development Searches. Box and whisker plots with the horizontal line denoting the median, X indicating the mean, and the bottom and top borders of the box representing the first and third quartiles. The whiskers extend to the minimum and maximum values while additional points indicate outliers that are greater than 1.5 times the interquartile range.

**Figure 2 behavsci-13-00429-f002:**
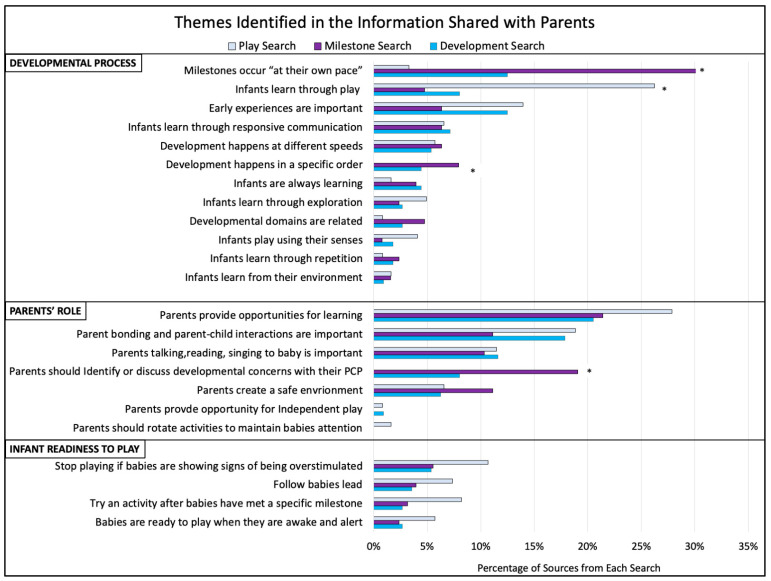
Themes identified in the information shared with parents. The percentage of sources from the Play, Milestone, and Development Searches that mentioned each theme identified regarding the developmental process, parents’ role in development, and infant readiness to play; themes are shown from highest to lowest rate of occurrence within these topic areas. * Denotes significant findings. PCP = Primary Care Provider.

## Data Availability

The data presented in this study are available in the Appendix A.

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
