# Peer review of "Information Available to Parents Seeking Education about Infant Play, Milestones, and Development from Popular Sources"

_behavsci, 2023, doi:10.3390/bs13050429_

Round 1

Reviewer 1 Report

General Comments and Overall Evaluation:

     I think this is an interesting study that examines a practical issue regarding parent resources on the internet and whether these resources provide up-to-date scientifically validated strategies that give parents an active versus passive role in promoting their infant’s development. In general, the manuscript is well-organized. There is a little repetition in the Introduction (details below) that could be reduced but a rationale and justification for the review was provided. The materials and methods section was clear, but there is information that needs to be added (details below). The results were easy to follow and the discussion provided a good interpretation of the findings, but there needs to be better specificity in the language to reduce misleading statements (details below).

Specific Comments

Abstract:

-p.1, line 15: The authors use the phrase, “The results found…”, which should be changed to “The results showed…”.  Only people can find things.

Introduction:

-p.1, line 32-35: This is a long sentence, one of many in the manuscript, where the addition of the semi-colon is not helpful. The information after the semi-colon should be a separate sentence or it could be taken out altogether.  I recommend that the authors review their manuscript for long sentences that should be split up.

-p.1, line 37: The use of the word “development” suggests that the authors are going to focus on the development of infant education programs, which cannot be done with their study. In addition, the development of infant education programs isn’t the goal of their study. This is misleading to the reader and should be removed. The first paragraph should focus on the study goal.

-p.2, line 54-55: There is repetitive information in this paragraph that needs to be cut out.  The author state that parental knowledge is positively related to parent-child interaction and infant development a number of times (e.g., earlier in the paragraph, near the end of the paragraph, at the end of the paragraph).  The mention at the end of the second paragraph should be deleted since it’s not necessary and reduces the flow. If the sentence is removed, there is a better transition to the next paragraph.

Materials and Methods:

A brief explanation of the way the Google search engine provides results with specific phrases (e.g., “How to Play with baby”) would be helpful to understand the search results. Specific details regarding the process of the search between October 2019 and March 2021 are also needed.

-p. 3, line 132: The authors mention an online survey, but there are no details regarding this survey or the reason for its use.  This needs to be added. 

-p. 3 & 4, Source selection: Additional rationale is needed regarding the choice of search phrases (i.e., why those phrases?), the decision to include the top 150 searches in the initial search, and the decision to include the top 50 results from each search conducted in November 2022.  Why were those numbers chosen?

Discussion:

Throughout the discussion, the authors must make it clear that their conclusions and interpretations are based on a content review of materials that were retrieved based on specific search terms. This means that general statements regarding parent resources should be removed because that’s misleading to the reader.  It’s unclear whether their findings can be generalized across all parent resources about infant development.

For example:

-p.9, line 385 and line 395: In the second paragraph the authors state that overall, parents are most likely to learn about milestones when looking for information about infant play, milestones, and development.  This isn’t exactly true.  The authors used specific search terms, which resulted in a particular set of results. Their statement needs to be changed to reflect this fact.  Unless the authors were exhaustive in their search, their statement is misleading.  Similarly, the last sentence in this paragraph generalizes to “educational materials available to parents”, but did the authors examine all the educational materials that are available?

-p.10, line 417: The authors make a general statement about healthcare professionals, but their results don’t apply to all healthcare professionals, only those who have developed content that can be found with a specific Google search.

-p10, line 432: The authors use the acronym “PCP”.  Acronyms reduce readability and should be avoided unless it used continuously throughout the manuscript.

-p.11, line 462: The authors indicate that professionals should direct parents to utilize different search terms to better direct them to information.  This is an important implication of their findings but what do they mean by “utilize different search terms”?  How should parents decide on or formulate their search terms? Also, there needs to be a detailed discussion regarding the importance of search terms.  This would provide a clear context for their statement. For example, were the search terms that the authors used effective?  Why or why not?

-p.11, lines 478-483: This is a very long sentence to process and should be split into at least two sentences.

-p.11, last few lines: The last two sentences do not seem realistically related to the goal of the authors’ study. Although experts should be developing better resources for parents and making them accessible, how do we encourage this?  What is the mechanism by which this can happen?  How long would it take?  The authors’ suggestion doesn’t change the type of resources that are already out there and being found by parents.  It seems to me that an important implication of the authors’ findings is that parents should be more critical about the resources they are finding regarding infant development and play. These resources could reflect old and outdated research, which limits their usefulness.

Author Response

Thank you for your review. Please see our responses to your comments attached.

Reviewer 2 Report

Dear author(s)

It was my pleasure to review your manuscript entitled “Information Available to Parents Seeking Education About Infant Play, Milestones, & Development from Popular Sources” and advise you to prosper your current research project. In my view, your topic has touched on a critical issue in a fascinating context. However, there are many spaces to be improved in terms of argumentation, theoretical background, research method, and findings. I hope my below comments would help you develop your work into groundbreaking research in your domain.

Much sharper problematization is required so that the introduction draws the reader into the paper. The introduction therefore needs to do a better job in setting the stage for the articulation of the theoretical contributions of the study. At the end of the introduction, we should have a clear idea of what the paper is about (i.e. its motivation, the gap in understanding that the paper is trying to address and summary of theoretical contributions).

 At the moment. the chapter is that is now entitled as "Conclusion" should link back to the literature and show theoretical contributions, that exceed the conclusion that some literature was "inline" with the findings of the authors.

The authors need to draw substantive conclusions from their results, and suggest, develop recommendations for further research.

- Using the following reference could be beneficial as these add more evidence to the literature review section:

 Studying the influence of emotional intelligence on the organizational innovation. International Journal of Human Capital in Urban Management, 3(1), 45-52. ‏ 10.22034/IJHCUM.2018.03.01.005.  (2018).

Best of luck with the further development of the paper.

 Another round of spellchecking by a native speaker is recommended.

Author Response

Thank you for your review. Please see our responses to your comments in the attached document.
